# 🛋 CRAFT: A Benchmark for Causal Reasoning About Forces and inTeractions

**Tayfun Ates**[1,†]**, Muhammed Samil Atesoglu**[1]**, Cagatay Yigit**[1]**, Ilker Kesen**[2]**, Mert Kobas**[3]**,
Erkut Erdem**[1]**, Aykut Erdem**[2]**, Tilbe Goksun**[3]**, Deniz Yuret**[2]

[1] Department of Computer Engineering, Hacettepe University, Ankara, Turkey
[2] Department of Computer Engineering, Koç University, Istanbul, Turkey
[3] Department of Psychology, Koç University, Istanbul, Turkey
[†]Correspondence: tates@hacettepe.edu.tr

## Abstract

Recent advances in Artificial Intelligence and deep learning have revived the interest in studying the gap between the reasoning capabilities of humans and machines. In this ongoing work, we introduce CRAFT, a new visual question answering dataset that requires causal reasoning about physical forces and object interactions. It contains 38K video and question pairs that are generated from 3K videos from 10 different virtual environments, containing different number of objects in motion that interact with each other. Two question categories from CRAFT include previously studied *descriptive* and *counterfactual* questions. Besides, inspired by the theory of force dynamics from the field of human cognitive psychology, we introduce new question categories that involve understanding the intentions of objects through the notions of *cause*, *enable*, and *prevent*. Our preliminary results demonstrate that even though these tasks are very intuitive for humans, the implemented baselines could not cope with the underlying challenges.

## 1 Introduction

The collection of abilities of humans to understand and make approximate predictions about physical environments consisting of various objects that are in steady state or in motion is known as *intuitive physics* [Kubricht et al., 2017]. Cognitive scientists have extensively studied the factors that affect infants' or adults' ability of physical reasoning [Baillargeon, 1995, 2008, Téglás et al., 2011, Battaglia et al., 2013]. Some of these abilities have also been studied for other animals such as chicks (Gallus gallus) [Chiandetti and Vallortigara, 2011]. Recent advances in machine learning have enabled computers to understand what type of object is present in a specified image (*classification)*, which bounding box best wraps that object (*detection)*, what its exact boundaries are (*segmentation)*. Although these artificial vision systems have shown astounding progress in the past decade, there are some areas in which these systems are still significantly below human performance. One such area includes the capability of humans to reason about physical actions of objects by observing their environment. This is a recent research direction for which cognitive and machine learning scientists are working together to bring similar capabilities to artificially intelligent robots so that they acquire similar intuitions and better understand their surroundings.

A crucial point that is worth mentioning here is that improving physical reasoning capabilities can also make agents better anticipate the results of their actions in their physical environments. They can gain the ability to consider counterfactual actions without actually performing them. They can estimate what will happen if they perform a specific action. One of the recent examples in this

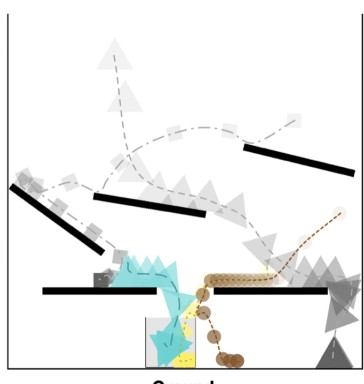

**Descriptive**
Q: "How many objects **fall to** the **ground**?" A: "2"
Q: "After **entering** the **basket**, does the **small yellow square** collide with other objects?" A: "True"
**Counterfactual**
Q: "How many objects **fall to** the **ground**, if the **small yellow box** is removed?" A: "1"
Q: "Does the **small gray box enter** the **basket**, if any other single one of the objects is removed?" A: "True"
**Cause**
Q: "Does the **small brown sphere cause** the **tiny yellow box** to **enter** the **basket**?" A: "True"
Q: "Does the **small gray block cause** the **big cyan triangle** to collide with the **basket**?" A: "True"
**Enable**
Q: "Does the **small brown sphere enable** the **small yellow box** to **enter** the **basket**?" A: "False"
Q: "How many objects does the **small** gray **block enable** to **enter** the **basket**?" A: "0"
**Prevent**
Q: "Does the **small yellow square prevent** the **tiny brown circle** from **entering** the **basket**?" A: "True"
Q: "How many objects does the **large cyan triangle prevent** from **entering** the **basket**?" A: "1"

Figure 1: Example CRAFT questions from a sample scene. There are 10 different scene layouts and 65 different question types which are divided into five distinct categories. Besides having tasks questioning descriptive attributes that possibly require temporal reasoning, CRAFT proposes new challenges including more complex tasks that need single or multiple counterfactual analysis, or understanding object intentions for deep causal reasoning.

direction is the Jenga-playing robot [Fazeli et al., 2019]. We believe intuitive physics is an essential ability to develop robots that are safe to interact with humans.

In this work, our main aim is to help machine learning models to understand and reason about physical relationships between dynamic objects in a scene. We propose a new visual question answering task, named CRAFT (Causal Reasoning About Forces and inTeractions), which requires understanding complex physical reasoning to be able to score high. CRAFT is designed to be complex for artificial models and simple for humans. Our dataset contains virtually generated videos of 2-dimensional scenes with accompanying questions. Its most prominent properties are that it contains visuals with complex physical interactions between objects and questions that test strong reasoning capabilities. For example, answering the questions typically require detecting objects, tracking their states in relation to other objects, which in turn can be attributed to causing, enabling or preventing certain events. Moreover, there are also counterfactual questions about understanding what would have happened if a slight change occurred in the environment. Figure 1 shows sample questions from CRAFT from 5 different categories, whose details are to be provided, for a single simulation [1].

Our main contribution is the creation of a novel dataset that uses language and vision to test spatiotemporal reasoning on complex physical systems. In addition, we experiment with some simple baselines and demonstrate that they are insufficient to handle the challenges CRAFT introduces. We hope that our work will lead to the generation of better systems on the path of approaching human intelligence for physical reasoning.

## 2   Related Work

**Visual Question Answering.** The datasets for visual question answering (VQA) can be categorized along two dimensions. The first dimension is the type of visuals, which include either real world images [Malinowski and Fritz, 2014, Ren et al., 2015, Antol et al., 2015, Zhu et al., 2016, Goyal et al., 2017] or videos [Tapaswi et al., 2016, Lei et al., 2018], or synthetically created content [Johnson et al., 2017, Zhang et al., 2016, Yi et al., 2020]). The second is at how the questions and answers are collected, which are usually done via crowdsourcing [Malinowski and Fritz, 2014, Antol et al., 2015] or by automatic means [Ren et al., 2015, Lin et al., 2014, Johnson et al., 2017]). An important challenge for creating a good VQA dataset lies in minimizing the dataset bias. A model may exploit such biases and cheat the task by learning some shortcuts. In our work, we generate questions about simulated scenes using a pre-defined set of templates by considering some heuristics to eliminate strong biases. As compared to the existing VQA datasets, our CRAFT dataset is specifically designed to test the agents' understanding of dynamic state changes of the objects in a scene. Although

---

[1]More examples from CRAFT can be found in appendix A.5 and `https://sites.google.com/view/craft-examples/home`

some existing VQA datasets question temporal reasoning capabilities of models [Lei et al., 2018, Yu et al., 2019, Lei et al., 2020], they do not require the models to have a deep understanding of intuitive physics to answer the questions, the only exceptions being What-If [Wagner et al., 2018] and CLEVRER [Yi et al., 2020]. In that sense, CRAFT shares a similar design goal with these datasets – however the scenes in our benchmark are more complex, as explained later.

**Intuitive Physics in Cognitive Science.** Common sense can be considered as humans' collection of capabilities to perceive, understand and judge everyday situations. Intuitive physics, an important part of commonsense knowledge, is related to people's perceptions of changes in physical world and their own understanding of how physical phenomena works [Proffitt and Kaiser, 2006]. Different theories have been proposed by cognitive scientists to model how humans learn, experience, and perform physical reasoning for certain events. Some of them are mental model theory [Khemlani et al., 2014], causal model theory [Sloman et al., 2009], and force dynamics theory [Wolff and Barbey, 2015], which try to represent a variety of causal relationships such as cause, enable, and prevent between two main entities, i.e. an affector and a patient (the object the affector acts on). To our knowledge, our work is the first attempt at integrating these complex causal relationships in a VQA setup for machine learning models to improve their physical reasoning capabilities.

**Intuitive Physics in Artificial Intelligence.** In recent years, there has been a growing interest within the AI community in developing models that have reasoning about intuitive physics. For instance, some researchers have explored the problem of predicting whether a set of objects are in stable configuration or not [Mottaghi et al., 2016] or if not where they fall [Lerer et al., 2016]. Others have tried to estimate a motion trajectory of a query object under different forces [Mottaghi et al., 2016] or developed methods to build a stack configuration of the objects from scratch through a planning algorithm [Janner et al., 2019]. Li et al. [2019] suggested to represent rigid bodies, fluids, and deformable objects as a collection of particles and used this representation to learn how to manipulate them. Very recently, Bakhtin et al. [2019] and Allen and Tenenbaum [2020] created the PHYRE and the Tools benchmarks, respectively, which both include different types of 2D-environments. An agent must reason about the scene and predict the outcomes of possible actions in order to solve the task associated with the environment. Although these works involve complicated physical reasoning tasks, the language component is largely missing. As we mentioned earlier, Wagner et al. [2018] and Yi et al. [2020] created VQA datasets for intuitive physics, but they lack visual variations unlike PHYRE and Tools. In that sense, our CRAFT dataset simply combines the best of both worlds. Moreover, in addition to the two types of questions investigated in CLEVRER [Yi et al., 2020], namely *descriptive* and *counterfactual*, CRAFT also involves questions that need reasoning about *cause*, *enable*, and *prevent*.

# 3 The CRAFT Dataset

CRAFT is built to evaluate temporal and causal reasoning capabilities of algorithms using videos of 2D simulations and related questions. Current version of CRAFT has nearly 38K video and question pairs that are automatically generated from 3K videos.

**Video Generation.** We use Box2D [Catto, 2010] to create our virtual scenes. There are 10 distinct scene layouts from which 10 seconds of videos are collected with a spatial resolution of 256 by 256 pixels.

**Objects.** There are static elements and dynamic objects in our scenes. Each scene includes variable number of and different types of those. Attributes of the static elements such as position or orientation are decided at the beginning of the simulation and they are fixed throughout the video sequence (see Figure 2). The values of these attributes are assigned from a set of random intervals which are predefined for each type of scene. Attributes of dynamic objects, on the other hand, are in continuous change throughout the sequence due to gravity or other interactions that they are subject to, until they rest. Furthermore, there are 6 static element types for our scenes, namely *(ramp, platform, basket, left wall, right wall, ground)*. Static elements are all drawn in *black* color in the video sequences. The set of the dynamic objects contains 3 shapes *(cube, triangle, circle)*, 2 sizes *(small, large)*, and 8 colors *(gray, red, blue, green, brown, purple, cyan, yellow)*.

**Events.** To represent the dynamism in the simulations formally, we detect different types of events from our simulations. They are *Start*, *End*, *Collision*, *Touch Start*, *Touch End*, *In Basket*. *Start* and *End* events represent the start and the end of the simulations, respectively. Although we question

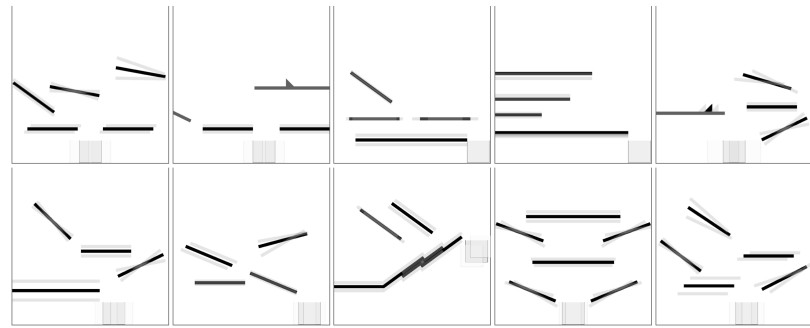

Figure 2: Variations within each scene layout. While generating the videos, the attributes of the static scene elements are sampled from a uniform distribution. Opaque illustrations demonstrate the mean positions whereas transparent drawings show the extreme cases.

*Collision* events only in our tasks, we want algorithms to differentiate a collision from a touching event. Therefore, *Touch Start*, *Touch End* are detected as well. Finally, *In Basket* event is triggered if the object enters the one and only basket in a scene. All events inside a simulation are represented in a causal graph for the question generator to extract causal relationships easily.

**Simulation Representation.** A video simulation sample is represented by 3 different structures, which are *scene representation at the start*, *scene representation at the end*, and *causal graph of events*, respectively. Scene representations hold information regarding the objects' static and dynamic attributes such as color, position, shape, velocity, etc. at the start and at the end of the simulation. These structures are described as collections of attributes of objects. On the other hand, the last structure holds causal relationships between events of a simulation. These structures have sufficient information to find the correct answers to CRAFT questions. Our simulation system also allows us to generate scene graphs like the ones used in CLEVR [Johnson et al., 2017], though we have not investigated it yet, which might be used for spatial reasoning.

**Question Generation.** Each CRAFT question is represented with a functional program as in CLEVR. We use a different set of functional modules for our programs extending the CLEVR approach. For example, our module set includes, but is not limited to functions which can filter events such as *In Basket* and *Collision*, and functions which can filter objects based on whether they are stationary at the start or the end of the video. List of our functional modules and some example programs are provided in appendices A.3 and A.4, respectively. Moreover, we use a different set of word synonyms and allow question text to be paraphrased for language variety similar to CLEVR.

**Tasks.** CRAFT has 65 different question types under 5 different categories which are *Descriptive, Counterfactual, Enable, Cause, Prevent*. *Descriptive* tasks mainly require extracting the attributes of objects and some of them, especially those involving counting, need some temporal analysis as well. CRAFT extends the work CLEVRER by Yi et al. [2020] with different types of events and multiple environments. *Counterfactual* tasks require understanding what would happen if one of the objects was removed from the scene. Exclusive to CRAFT, some *Counterfactual* tasks (*"Does the small gray circle enter the basket, if any other single one of the objects is removed?"*) require multiple counterfactual simulations to be answered. As an extension to *Counterfactual* tasks, *Enable, Cause, Prevent* tasks require grasping what is happening inside both the original video and the counterfactual video. In other words, models must infer whether an object is causing or enabling an event or preventing it by comparing the input video and the counterfactual video that should be simulated somehow. In the question text, the affector and the patient objects are explicitly specified. Some of these tasks include multiple patients.

Finally, in order to have a better understanding of the differences between *Enable*, *Cause*, and *Prevent* questions, one should understand the "intention" of the objects. We identify the intention in a simulation by examining the initial linear velocity of the corresponding object. If the magnitude of the velocity is greater than zero, then the object is intended to do the task specified in the question text, such as entering the basket or colliding with the ground. If the magnitude of the velocity is zero, then the object is not intended to do the task, even if there is an external force, such as gravity, upon it at the start of the simulation. Therefore, an affector can only enable a patient to do the task if the

Table 1: Performance of all baseline models on CRAFT training, validation and test splits. We report the average accuracy. C, CF, D, E and P columns stand for *Cause*, *Counterfactual*, *Descriptive*, *Enable* and *Prevent* tasks, respectively.

| Baseline | Input | Train | Validation | Test | | | | | |
|---|---|---|---|---|---|---|---|---|---|
| | | | | C | CF | D | E | P | All |
| MFA | Question | 28.61 | 27.86 | 30.50 | 42.36 | 21.54 | 27.31 | 27.54 | 28.00 |
| AT-MFA | Question | 42.83 | 41.29 | 45.60 | 47.11 | 38.14 | 47.60 | 44.59 | 41.48 |
| LSTM | Question | 58.73 | 44.76 | 53.77 | 52.99 | 39.16 | 52.77 | 55.08 | 44.65 |
| LSTM-CNN | Question + First Frame | 93.14 | 47.68 | 44.34 | 50.34 | 46.32 | 52.03 | 53.77 | 47.83 |
| LSTM-CNN | Question + Last Frame | 90.94 | 53.34 | 50.63 | 56.71 | 53.99 | 52.40 | 51.48 | 54.42 |
| Human | Question + Video | – | – | 66.67 | 74.07 | 93.01 | 59.09 | 90.9 | 85.89 |

patient is intended to do it but fails without the affector. Similarly, an affector can only cause a patient to do the task if the patient is not intended to do it. Moreover, an affector can only prevent a patient from doing the task if the patient is intended to do it and succeeds without the affector.

**Reducing Dataset Bias.** CRAFT contains simulations from different scenes increasing the variety in the visual domain as well. This variety also makes minimizing the dataset biases difficult because of the multiplicity in the number of the domains (textual and visual). Our data generation process enforces different simulation and task pairs to have uniform answer distributions while trying to keep overall answer distribution as uniform as possible. Appendix A.1 provides some information about dataset statistics.

## 4 Experimental Analysis

### 4.1 Baselines

We implemented four simple baselines to find out how much they are able to achieve on CRAFT. **Most Frequent Answer** model (MFA) uses the heuristic of finding the most frequent answer in the training split, and then outputting that answer for all questions. **Answer type-based Most Frequent Answer** model (AT-MFA) exploits the answer type (e.g. color, shape, boolean), finds the most frequent answers for each type of question in the training split, and then, outputs those answers by exploiting answer types. **LSTM** model is image-blind that it processes the question with an Long Short-term Memory network (LSTM) [Hochreiter and Schmidhuber, 1997], then predicts an answer without using the visual input. **LSTM-CNN** model processes the question with an LSTM and the frame(s) with Residual Networks (ResNet) [He et al., 2016]. It integrates these modalities by concatenating the extracted features, and then predicts an answer accordingly. Training and implementation details of the neural models are presented in appendix A.2.

### 4.2 Results

We use the accuracy metric to evaluate the baseline models. Table 1 shows the performances of the baselines on all dataset splits, including task-specific performances on the test split. Simulation column denotes the inputs used by the baselines to represent the simulation video. We trained two different models for LSTM-CNN baseline. While the first one employs the first frame of the simulation as visual input, the second one utilizes the last frame of the simulation. We also collected data from 12 adults to respond to our questions (522 questions) after watching the videos. People responded to 489 questions and among them, we received an average of 85 percent correct responses. *Cause* and *Enable* categories received the least correct responses.

## 5 Discussion and Future Work

In this work, we present CRAFT, a new benchmark to challenge intuitive physics capabilities of the current machine learning algorithms. Besides providing information about our efforts for creating this dataset, we provide some very simple baseline results. Our preliminary results demonstrate that the tasks about physical interactions between objects that seem intuitive to humans might be hard

for machines. There is a large gap ($> 30\%$) between humans and our neural baselines, though we must admit that these are relatively simple baselines. In addition to the performance analysis of artificial models, detailed studies on human subjects solving CRAFT tasks are also required in order to understand differences between humans and machines.

There may be some extensions of CRAFT which can be considered as future work. For example, current version of CRAFT includes multiple patients in cause, enable, and prevent tasks, but does not include multiple affectors. Moreover, other static object attributes, such as density, can be integrated using material textures. Finally, our programs of tasks depend on the end results of the simulations to be able to provide correct answers to the questions. Investigation of temporally local relationships between objects may also be interesting. We believe that improving CRAFT and algorithms trying to solve CRAFT is a new research direction for artificial intelligence systems mimicking humans for causal reasoning about forces and interactions. We are actively working on extending CRAFT, and welcome collaboration.

## Broader Impact

This work does not present any foreseeable societal consequence.

## Acknowledgments

This work was supported in part by an AI Fellowship to Ilker Kesen provided by the KUIS AI Lab and TUBA GEBIP fellowship awarded to E. Erdem.

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

# A   Appendix

## A.1   Dataset Statistics

To account for bias effects, we generated our question answer pairs in a controlled manner so that the frequency of the answers are almost balanced over our 10 different scene layouts and 65 different question types. CRAFT has currently over 38K video and question pairs, which are generated from 3K videos in an automatic manner. Here, our aim is to make it harder for the models to find simple shortcuts by predicting the task identifier, the simulation identifier, or both, instead of understanding the scene dynamics and the question. Figure A.1 shows the answer distributions for the task categories in CRAFT.

## A.2   Training and Implementation Details

Our neural baselines encode the question text by using an LSTM network having 256 hidden units with randomly initialized word-vector embeddings. The words in the question are processed sequentially, and the last hidden state of the LSTM network are used as the final textual representation. LSTM-CNN baselines uses the output of the fourth convolutional downsampling layer of pretrained ResNet-18 [He et al., 2016] to encode the visual information. The flattened visual representations and the textual representations are concatenated, then is processed with a linear layer containing 256 units followed by a $tanh$ activation, and finally a linear layer generates the answer scores. We also applied dropout [Srivastava et al., 2014] to visual and textual representations with 0.2 probability.

We use Adam optimizer [Kingma and Ba, 2014] with learning rate $\alpha = 0.0001$. We train all models for 30 epochs on Tesla K80 and Tesla T4 GPUs. Each epoch takes at most one and half hour depending on the baseline. For each baseline, we report the accuracies considering the model which achieves the best performance on the validation split. We use floating point pixels values, between 0 and 1. We initialize ResNet-18 model pretrained on ImageNet 2012 dataset [Deng et al., 2009].

## A.3   Functional Modules

CRAFT questions are represented with functional programs. Input and output types for our functional modules are listed in Table A.1. Lists of all functional modules are also provided in Tables A.2-A.6.

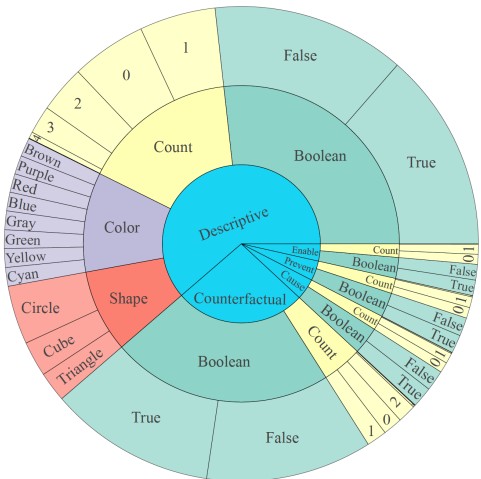

Figure A.1: Statistics of the questions in CRAFT dataset. Innermost layer represents the distribution of the questions for different task categories. Middle layer illustrates the distribution of the answer types for each task category. Outermost layer represents the distribution of answers for each answer type.

Table A.1: Input and output types of functional modules in CRAFT.

| Type | Description |
|---|---|
| *Object* | A dictionary holding static and dynamic attributes of an object |
| *ObjectSet* | A list of unique objects |
| *ObjectSetList* | A list of *ObjectSet* |
| *Event* | A dictionary holding information of a specific event |
| *EventSet* | A list of unique events |
| *EventSetList* | A list of *EventSet* |
| *Size* | A tag indicating the size of an object |
| *Color* | A tag indicating the color of an object |
| *Shape* | A tag indicating the shape of an object |
| *Integer* | Standard integer type |
| *Bool* | Standard boolean type |
| *BoolList* | A list of *Bool* |

## A.4 Example Programs

In this, we provide example functional programs for some of the sample questions provided in Figure 1, which are used to extract the correct answers using our simulation environment. Figures A.2 to A.6 provide functional program samples that are designed for CRAFT descriptive, counterfactual, cause, enable, and prevent questions, respectively.

## A.5 Additional Examples

Figures A.7 and A.8 provide other CRAFT examples and corresponding predictions by our baseline models.

Table A.2: Input functional modules in CRAFT.

| Module | Description | Input Types | Output Type |
|---|---|---|---|
| SceneAtStart | Returns the attributes of all objects at the start of the simulation | *None* | *ObjectSet* |
| SceneAtEnd | Returns the atttributes of all objects at the end of the simulation | *None* | *ObjectSet* |
| StartSceneStep | Returns 0 | *None* | *Integer* |
| EndSceneStep | Returns -1 | *None* | *Integer* |
| Events | Returns all of the events happening between the start and the end of the simulation | *None* | *EventSet* |

Table A.3: Output functional modules in CRAFT.

| Module | Description | Input Types | Output Type |
|---|---|---|---|
| QueryColor | Returns the color of the input object | *Object* | *Color* |
| QueryShape | Returns the shape of the input object | *Object* | *Shape* |
| Count | Returns the size of the input list | *ObjectSet* | *Integer* |
| Exist | Returns true if the input list is not empty | *ObjectSet / EventSet* | *Bool* |
| AnyFalse | Returns true if there is at least one false in a boolean list | *BoolList* | *Bool* |
| AnyTrue | Returns true if there is at least one true in a boolean list | *BoolList* | *Bool* |

Table A.4: Object filter functional modules in CRAFT.

| Module | Description | Input Types | Output Type |
|---|---|---|---|
| FilterColor | Returns the list of objects which have a color same with the input color | *(ObjectSet, Color)* | *ObjectSet* |
| FilterShape | Returns the list ofobjects which have a shape same with the input shape | *(ObjectSet, Shape)* | *ObjectSet* |
| FilterSize | Returns the list of objects which have a size same with the input size | *(ObjectSet, Size)* | *ObjectSet* |
| FilterDynamic | Returns the list of dynamic objects from an object set | *ObjectSet* | *ObjectSet* |
| FilterMoving | Returns the list of objects that are in motion at the step specified | *(ObjectSet, Integer)* | *ObjectSet* |
| FilterStationary | Returns the list of objects that are stationary at the step specified | *(ObjectSet, Integer)* | *ObjectSet* |

Table A.5: Event filter functional modules in CRAFT.

| Module | Description | Input Types | Output Type |
|---|---|---|---|
| FilterEvents | Returns the list of events about a specific object from an event set | *(EventSet, Object)* | *EventSet* |
| FilterCollision | Returns the list of collision events from an event set | *EventSet* | *EventSet* |
| FilterCollisionWithDynamics | Returns the list of collision events involving dynamic objects | *EventSet* | *EventSet* |
| FilterCollideGround | Returns the list of collision events involving the ground | *EventSet* | *EventSet* |
| FilterCollideGroundList | Returns the list of collision event sets involving the ground | *EventSetList* | *EventSetList* |
| FilterCollideBasket | Returns the list of collision events involving the basket | *EventSet* | *EventSet* |
| FilterCollideBasketList | Returns the list of collision event sets involving the basket | *EventSetList* | *EventSetList* |
| FilterEnterBasket | Returns the In Basket events | *EventSet* | *EventSet* |
| FilterEnterBasketList | Returns the list of In Basket event sets | *EventSetList* | *EventSetList* |
| FilterBefore | Returns the events from the input list that happens before input event | *(EventSet, Event)* | *EventSet* |
| FilterAfter | Returns the events from the input list that happened after input event | *(EventSet, Event)* | *EventSet* |
| FilterFirst | Returns the first event | *EventSet* | *Event* |
| FilterLast | Returns the last event | *EventSet* | *Event* |
| EventPartner | Returns the object interacting with the input object through the specified event | *(Event, Object)* | *Object* |
| FilterObjectsFromEvents | Returns the objects from the specified events | *EventSet* | *ObjectSet* |
| FilterObjectsFromEventsList | Returns the list of object sets from a list of event sets | *EventSetList* | *ObjectSetList* |
| GetCounterfactEvents | Returns the event list if a specific object is removed from the scene | *Object* | *EventSet* |
| GetCounterfactEventsList | Returns the counterfactual event list for all objects in an object set | *ObjectSet* | *EventSetList* |

Table A.6: Auxiliary functional modules in CRAFT.

| Module | Description | Input Types | Output Type |
|--------|-------------|-------------|-------------|
| Unique | Returns the single object from the input list, if the list has multiple elements returns INVALID | *ObjectSet* | *Object* |
| Intersect | Applies the set intersection operation | *(ObjectSet, ObjectSet)* | *ObjectSet* |
| IntersectList | Intersects an object set with multiple object sets | *(ObjectSetList, ObjectSet)* | *ObjectSetList* |
| Difference | Applies the set difference operation | *(ObjectSet, ObjectSet)* | *ObjectSet* |
| ExistList | Applies the Exist operation to each item in the input list returning a boolean list | *ObjectSetList / EventSetList* | *BoolList* |
| AsList | Returns an object set containing a single element specified by the input object | *Object* | *ObjectSet* |

**Question**: *"How many objects fall to the ground?"*

```
Count (
        FilterDynamic (
                FilterObjectsFromEvents (
                        FilterCollideGround (
                                Events ()
                        )
                )
        )
)
```

**Question**: *"After entering the basket, does the small yellow square collide with other objects?"*

```
Var QueryObject = FilterShape ( FilterColor ( FilterSize ( SceneAtStart(), "Small" ) , "Yellow"), "Cube" )
Var SmallYellowCubeEvents = FilterEvents ( Events(), QueryObject )
Exist (
        FilterAfter (
                FilterCollisionWithDynamics ( SmallYellowCubeEvents ),
                        FilterFirst (
                                FilterEnterBasket ( SmallYellowCubeEvents )
                        )
                )
        )
)
```

Figure A.2: Example programs for *descriptive* questions.

**Question**: *"How many objects fall to the ground, if the small yellow box is removed?"*

```
Var QueryObject = FilterShape ( FilterColor ( FilterSize ( SceneAtStart(), "Small" ) , "Yellow"), "Cube" )
Count (
        FilterObjectsFromEvents (
                FilterCollideGround (
                        GetCounterfactEvents ( QueryObject )
                )
        )
)
```

**Question**: *"Does the small gray box enter the basket, if any other single one of the objects is removed?"*

```
Var QueryObject = FilterShape ( FilterColor ( FilterSize ( SceneAtStart(), "Small" ) , "Gray"), "Cube" )
Var OtherDynamicObjects = Difference ( FilterDynamic ( SceneAtStart() ), AsList ( QueryObject ) )
AnyTrue (
        ExistList (
                IntersectList (
                        FilterObjectsFromEventsList (
                                FilterEnterBasketList (
                                        GetCounterfactEventsList ( OtherDynamicObjects )
                                )
                        ),
                        AsList (
                                QueryObject
                        )
                )
        )
)
```

Figure A.3: Example programs for *counterfactual* questions.

**Question**: *"Does the small brown sphere cause the tiny yellow box to enter the basket?"*

```
Var AffectorObject = FilterShape ( FilterColor ( FilterSize ( SceneAtStart(), "Small" ) , "Brown"), ''Circle'' )
Var PatientObject = FilterShape ( FilterColor ( FilterSize ( SceneAtStart(), "Small" ) , "Yellow"), "Cube" )
Exist (
        FilterStationary (
                Intersect (
                        Difference (
                                FilterObjectsFromEvents (
                                        FilterEnterBasket (
                                                Events()
                                        )
                                ),
                                FilterObjectsFromEvents (
                                        FilterEnterBasket (
                                                GetCounterfactEvents (
                                                        AffectorObject
                                                )
                                        )
                                )
                        ),
                        AsList ( PatientObject )
                ),
                StartSceneStep()
        )
)
```

Figure A.4: Example program for *cause* questions.

**Question**: *"How many objects does the small gray block enable to enter the basket?"*

```
Var AffectorObject = FilterShape ( FilterColor ( FilterSize ( SceneAtStart(), "Small" ) , "Gray"), "Cube" )
Count (
        FilterMoving (
                Difference (
                        Difference (
                                FilterObjectsFromEvents (
                                        FilterEnterBasket (
                                                Events()
                                        )
                                ),
                                FilterObjectsFromEvents (
                                        FilterEnterBasket (
                                                GetCounterfactEvents (
                                                        AffectorObject
                                                )
                                        )
                                )
                        ),
                        AsList ( AffectorObject )
                ),
                StartSceneStep()
        )
)
```

Figure A.5: Example program for *enable* questions.

**Question**: *"Does the small yellow square prevent the tiny brown circle from entering the basket?"*

```
Var AffectorObject = FilterShape ( FilterColor ( FilterSize ( SceneAtStart(), "Small" ) , "Yellow"), "Cube" )
Var PatientObject = FilterShape ( FilterColor ( FilterSize ( SceneAtStart(), "Small" ) , "Brown"), "Circle" )
Exist (
        FilterMoving (
                Intersect (
                        Difference (
                                FilterObjectsFromEvents (
                                        FilterEnterBasket (
                                                GetCounterfactEvents (
                                                        AffectorObject
                                                )
                                        )
                                ),
                                FilterObjectsFromEvents (
                                        FilterEnterBasket (
                                                Events()
                                        )
                                )
                        ),
                        AsList ( PatientObject )
                ),
                StartSceneStep()
        )
)
```

Figure A.6: Example program for *prevent* questions.

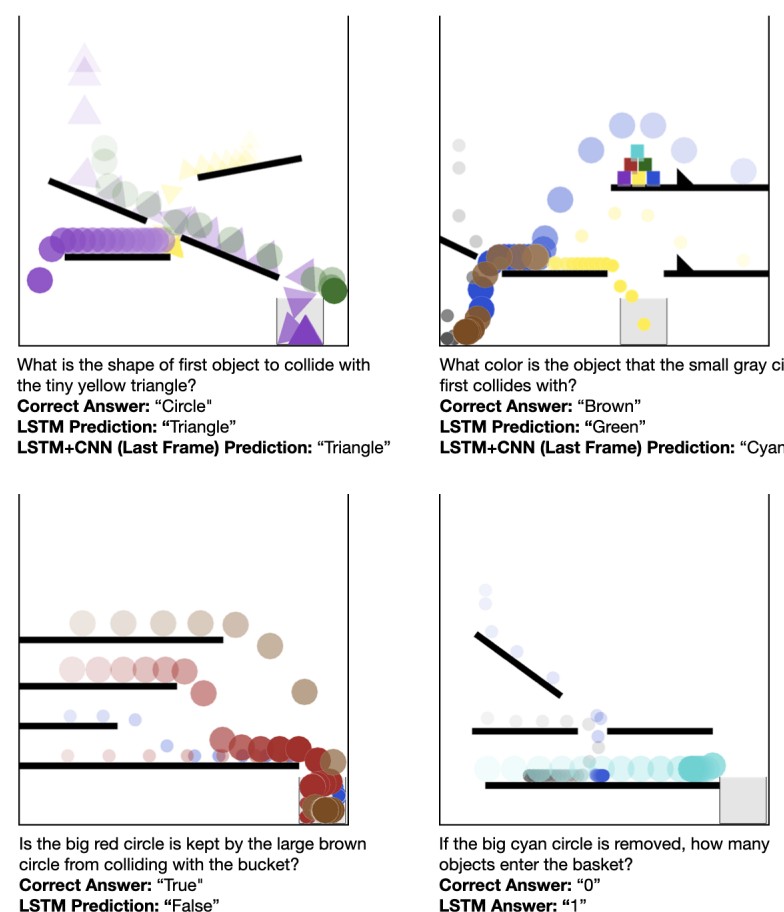

Figure A.7: Example model predictions. Upper row shows the cases that LSTM-CNN (First Frame) can correctly find the answer, whereas other models cannot. Similarly, lower row shows the cases of correct predictions by LSTM-CNN (Last Frame).

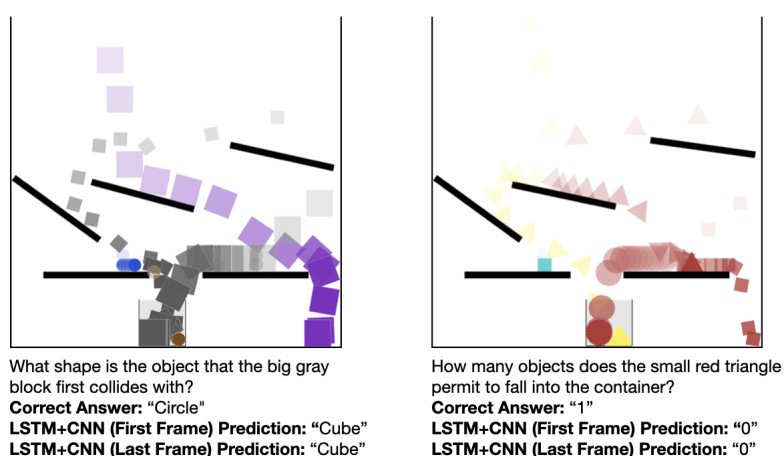

Figure A.8: Example model predictions showing the cases that LSTM can correctly find the answer whereas other models cannot.

