# OpenReview forum: "CRAFT: A Benchmark for Causal Reasoning About Forces and inTeractions"
_NeurIPS.cc/2020/Workshop/SVRHM — SVRHM@NeurIPS Poster_

### Official Review · AnonReviewer3 · 2020-10-28

**Rating:** 7
**Confidence:** 5

**Review:**

Summary and Contributions: The authors present a new question-answering dataset that specifically targets intuitive physical reasoning capabilities. The benchmark contains descriptive questions which depend on the input video, counterfactual questions which depend on the agent to construct a counterfactual video, and intentional questions (that ask about causes/enablement/prevention) that require comparing the input video to the counterfactual video.

Strengths: this benchmark seems to capture many intuitive physical reasoning phenomena

Weaknesses:
- there are no human experiments so it is not definitive whether the tasks actually are easy for humans (which is part of the paper's main claim)
- the baselines presented are necessary, but also rather weak compared to the state of the art machine learning models. It is not clear to me how much trouble Interaction Networks [1], Neural Physics Engine [2], or object-centric world models like OP3 [3], G-SWM [4] would have in these tasks.

Recommendations:
- This is an exciting benchmark and I hope to see the questions posed in this benchmark studied more in the AI community
- Add human experiments to justify claim that the benchmark is indeed easy for humans
- Compare against state-of-the-art machine learning models to highlight the gap between what further machine learning research needs to be done.

[1] Battaglia, P., Pascanu, R., Lai, M., & Rezende, D. J. (2016). Interaction networks for learning about objects, relations and physics. In Advances in neural information processing systems (pp. 4502-4510).
[2] Chang, M. B., Ullman, T., Torralba, A., & Tenenbaum, J. B. (2016). A compositional object-based approach to learning physical dynamics. arXiv preprint arXiv:1612.00341.
[3] Veerapaneni, R., Co-Reyes, J. D., Chang, M., Janner, M., Finn, C., Wu, J., ... & Levine, S. (2020, May). Entity abstraction in visual model-based reinforcement learning. In Conference on Robot Learning (pp. 1439-1456). PMLR.
[4] Lin, Z., Wu, Y. F., Peri, S., Fu, B., Jiang, J., & Ahn, S. (2020). Improving Generative Imagination in Object-Centric World Models. arXiv preprint arXiv:2010.02054.

---

### Official Review · AnonReviewer2 · 2020-10-29
**The paper proposes a dataset for studying dynamic physics understanding and causal reasoning with simple baseline models**

**Rating:** 7
**Confidence:** 4

**Review:**

* **Quality:** The dataset and tasks are well-designed. The research direction is promising. Connections to biology (thus to the theme of the workshop) could be further detailed.

* **Clarity:** The paper is well-written and clear.

* **Originality:** There are a number of similar datasets on intuitive physics. Yet this dataset features questions about causal interactions and reasoning, which is an interesting direction. However, I personally do not think it is a much more significant upgrade from the CLEVRER dataset. Another downside is that the objects do not look as real as what is in the CLEVRER dataset.

* **Significance:** The main contribution is the video dataset and tasks, which are reasonably novel. The work would be more significant if better baseline models or more detailed analyses of results are provided.

---

### Official Review · AnonReviewer1 · 2020-10-30
**Good dataset but missing human behavioral benchmarks**

**Rating:** 7
**Confidence:** 4

**Review:**

The paper presents a new dataset aimed at benchmarking visual question answering capabilities of machines on situations that require understanding of physical forces and object interactions. Specifically, the new dataset contains thousands of video and question pairs generated from 10 different virtual 2D environments. I appreciate the authors for recognizing the importance of causal and intuitive physical reasoning in human scene understanding and developing this dataset to push machine vision/language research towards developing more human-like abilities. This work extends some of the previous works (esp CLEVERER) by adding three new questions types for reasoning about cause, enable and prevent.

My biggest concern in reviewing this was that I had access to very few actual examples from the companion webpage and I could not get a clear sense of how difficult the task is for humans. If the goal is to push machines towards human capabilities, it is necessary to know how good humans are on this dataset. In light of this, I would like to see some benchmarks on human behavior for at least a subset of the dataset.

Minor comments:
1. I could not understand the difference between collision events and 'Touch start'/'Touch end' events, and felt the explanation lacking.
2. It feels like different objects have different material properties (for example, the big yellow ball is less 'bouncier' than the small green one). Is that intentional?

---

### Public Comment · ~Tayfun_Ates1 · 2020-12-07
**Update for the Camera-Ready Version**

We would like to thank all three reviewers for their valuable feedback and suggestions for future improvements.
Regarding the common concern about the lack of human experiments (R1 and R3), we have performed a user study on a small subset of the test split (having a similar distribution of question type/answer pairs). Our analysis shows that there is a large gap between our neural baselines and human subjects in that human subjects can achieve an overall accuracy of 85%. That being said, we will extend our analysis to better understand the differences between humans and machines.
Moreover, we plan to test the state-of-the-art object-centric models on CRAFT. As suggested by R3, these models might be better suited to the tasks in CRAFT.

---

### Decision · Program_Chairs · 2020-11-02

Accept (Poster)